# N₂O Emissions from Saline Soils in Response to Organic–Inorganic Fertilizer Application under Subsurface Drainage

**Yaming Zhai** [1] **, Qinyuan Zhu** [2] **, Ying Xiao** [3] **, Jingnan Chen** [4] **, Maomao Hou** [3] **and Lin Zhu** [2,*]

[1] College of Agricultural Science and Engineering, Hohai University, Nanjing 210098, China; njzhaiyaming@126.com
[2] Nanjing Institute of Environmental Sciences, Nanjing 210000, China; zhuqinyuan569@163.com
[3] College of Horticulture, Fujian Agriculture and Forest University, Fuzhou 350000, China; xiaoying0112@126.com (Y.X.); njhoumaomao@126.com (M.H.)
[4] College of Horticulture and Gardening, Fujian Agricultural Vocational and Technical College, Fuzhou 350000, China; 53cjn@sina.com
* Correspondence: zhulin@nies.org

**Abstract:** Organic fertilizer applications and subsurface drainage are two important measures for improving coastal saline soil; however, nitrous oxide (N₂O) emissions from saline soil under a combination of these two measures are seldom evaluated. In this study, saline soil cultivated with sunflowers (*Helianthus annuus* L.) was employed as an experimental system. Prior to the experiment, the saline soils were buried with three different spacings (10 m (S1), 15 m (S2), and 20 m (S3)) of subsurface drainage pipes. The nitrogen nutrients that are needed by sunflowers came from two different nitrogen sources (organic and inorganic fertilizer), including six application schemes of either 100% organic fertilizer (100%OF), 75% organic fertilizer combined with 25% inorganic fertilizer (75%OF), 50% organic fertilizer (50%OF), 25% organic fertilizer (25%OF), 0% organic fertilizer (0%OF), and no fertilizer (CK). The results show that the cumulative N₂O emissions from the treatments under S1, S2, and S3 throughout the entire growth period were 8.9–15.8, 9.5–17.5, and 10.1–17.6 kg ha$^{-1}$, respectively. A smaller spacing between adjacent drainage pipes or a higher replacement proportion of organic fertilizer reduced the accumulative N₂O emissions. The increased replacement of organic fertilizer decreased the soil salinity, whereas it increased the C/N ratio and total carbon content. The fertilization treatments significantly increased the nitrogen uptake of sunflower plants, with increase ranges of 18.1–47.2%, 8.6–40.5%, and 8.8–34.5% under S1, S2, and S3, respectively, compared with CK. The highest yield of sunflowers was achieved under S2 combined with 25%OF, reaching 3.82 t ha$^{-1}$. Correlation analysis showed that the N₂O emission flux was positively correlated with the soil salinity, crop yield, and crop nitrogen uptake, whereas it was negatively correlated with the total carbon, C/N ratio, and organic carbon content. We concluded that using 25% organic fertilizer instead of inorganic fertilizer was beneficial for reducing N₂O emissions while maintaining the crop yield under subsurface drainage.

**Keywords:** nitrogen oxide; nitrogen fertilizer; water management; coastal soil; sunflower

## 1. Introduction

Nitrous oxide (N₂O) is one of the main greenhouse gases that have attracted international attention. N₂O can generate NO radicals in the stratosphere and destroy the ozone layer by reacting with O₃ [1]. Compared with CO₂, N₂O increases the global warming efficiency 206-fold, and the annual concentration of N₂O in the troposphere increases by 0.26%, which inevitably has a huge impact on the global environment [2]. Among the numerous sources of N₂O emissions, arable land cannot be ignored. The annual N₂O emissions from arable land reach up to $6.4 \times 10^{12}$ g (calculated as pure nitrogen), accounting for 25% of the total emissions [3].

Due to the contradiction between urban land and agricultural land, developing coastal land resources has become a common choice for many countries and regions [4]. Coastal soil is formed by marine sediments under the movements of ocean tides or high concentrations of groundwater. Therefore, some coastal soils in certain regions have advantages such as a high organic matter content and abundant mineral nutrients. However, coastal soil has a high salt concentration, compact structure, high sand particle content, low available oxygen, and nutrient deficiency, often requiring improvement before planting crops [5]. The cultivation of salt-absorbing plants, organic fertilizer application, and subsurface drainage are the main means for improving coastal saline soils [6,7]. Sunflowers are a crop with only a moderate salt tolerance, but it is widely planted in coastal areas in order to improve saline soil due to its absorption ability. Research has shown that it can absorb excessive sodium ions, chloride ions, and other harmful salts in the soil, thereby alleviating salt stress for other plants [8]. In addition, sunflower roots can secrete acidic substances and convert carbonates into carbonate ions, further reducing soil salinity [9]. Organic fertilizer can positively change the soil structure and nutrient status, increase the number and activity of beneficial microorganisms, and improve the fertility of the soil by inputting a large amount of organic matter. Subsurface drainage can effectively lower the groundwater level, alleviate the salt accumulation, and improve the soil permeability and water use efficiency, thus promoting plant growth. A study [10] has shown that subsurface drainage can desalinate 15.4–62.2% of a 0–60 cm soil layer. At present, many studies [9,11] have attempted to improve coastal saline soil by combining the uses of plant engineering, water conservancy engineering, and agronomic measures. The effects of using different methods to improve soil properties and crop yield and quality have been widely studied.

In China, for many years, unreasonable fertilization methods and water management measures have disturbed the dynamic equilibrium of water and soil, significantly promoting $N_2O$ emissions. Researchers are constantly exploring pathways to reduce $N_2O$ emissions from arable lands. $N_2O$ emissions have been proven to be closely related to nitrogen fertilizer input. Shao [12] found that in a crop rotation system, when 25–50% organic fertilizer was used instead of the chemical fertilizer, the $N_2O$ emissions were clearly reduced while maintaining crop production. The experiments of Bi and Hao, on the leaves of fruits and vegetables, respectively, deduced that $N_2O$ emissions could be reduced by more than 50% through a reasonable organic and inorganic fertilizer application ratio [13,14]. Lu noted that compared with farmers who were accustomed to applying inorganic fertilizer nitrogen, reducing the amount of nitrogen fertilizer and combining it with nitrification inhibitors could significantly reduce $N_2O$ emissions; moreover, the nitrogen application rate of 400 kg ha$^{-1}$ combined with 2.0 kg ha$^{-1}$ of nitrapyrin obtained optimal effects regarding the reduction in $N_2O$ emissions [15]. Shi's research showed that the application of organic fertilizer promotes $N_2O$ emissions [16]. Liu found that the combined use of 75% pig manure and 25% inorganic fertilizer increases $N_2O$ emissions [17]. In summary, there are some differences among the previous studies on how different fertilizers affect $N_2O$ emissions, and the mechanisms need to be further studied.

Although many studies have evaluated the impact of using organic fertilizer to replace chemical fertilizers on $N_2O$ emissions, most of them were conducted in the inland field. For coastal land, which is a relatively new, and special agricultural land, whether there are differences in the regularity of $N_2O$ emissions, how $N_2O$ emissions are affected by saline soil agronomic measures such as salt-absorbing crop planting, subsurface drainage, as well as organic fertilizer application, are not clear. It is worth studying whether there is an optimal coupling mode for water conservancy measures, plant cultivation, and fertilizer application that can ensure crop output while reducing $N_2O$ emissions. This study employs saline soil cultivated with sunflowers (*Helianthus annuus* L.) as an experimental system. Before the experiment, the saline soils were treated with different spacings of subsurface drainage pipes, and the sunflowers were planted with different proportions of organic and inorganic fertilizers. The soil chemical indicators and plant growth indicators were observed. The objectives were: (1) to clarify the impact of subsurface spacing on $N_2O$

emissions to provide useful reference for optimizing the layout of drainage pipes; (2) to compare the differences in the $N_2O$ emission process and amount among the different ratios of organic and inorganic fertilizer application, for providing basis for optimizing fertilization strategy; and (3) to find out the main factors affecting $N_2O$ emission, in order to provide a theoretical foundation for the ecological management of coastal saline soil.

## 2. Materials and Methods

### 2.1. Experimental Site

The experiment was carried out in Tiaozi Mud Reclamation Area, Dongtai City, Yancheng, Jiangsu Province of China, from 8 June to 10 October 2021. The experimental site belongs to the boundary of subtropical and warm temperate zones, which have a significant monsoon climate and abundant rainfall. The experimental site has four distinct seasons, with an average annual rainfall of 1054.2 mm and sunshine hours of 2130.5 from 2010 to 2020. In recent years, the extreme highest temperature was 38.6 °C and the extreme lowest temperature was −5.8 °C. The soil properties (0–20 cm) were: a total salt content of 4.2 g kg$^{-1}$, organic matter content of 2.1%, available nitrogen of 114.6 mg kg$^{-1}$, available phosphorus of 8.8 mg kg$^{-1}$, and available potassium of 110.7 mg kg$^{-1}$, respectively. The experimental field was planted with reeds for three years.

### 2.2. Experimental Design

The experiment employed the sunflower variety "Baikuiza 6" (*Helianthus annuus* L.) as the plant material; their seeds were sown on 8 June. The division of the growth period is shown in Table 1. According to the local cultivation habits, the municipal waters with 0.3 mg L$^{-1}$ of nitrogen content and 0.6 ms cm$^{-1}$ of EC were used for irrigation, with irrigation amounts of 76.6 mm, 82.5 mm, 74.1 mm, and 62.6 mm on 15 July, 4 August, 19 August, and 16 September, respectively. The furrow irrigation method was employed. The cumulative rainfall during the experimental period was 92.1 mm. Therefore, the total amount of water obtained by the sunflowers was 387.9 mm. All subsurface drainage pipes in the experiment were buried at a depth of 1.2 m.

**Table 1.** Division of sunflower growth stages.

|  | Seedling Stage | Budding Stage | Blooming Stage | Harvest Stage |
| --- | --- | --- | --- | --- |
| Growing stage | 8 June–21 July | 22 July–15 August | 16 August–30 August | 31 August–10 October |
| Irrigation amount | 76.6 mm | 82.5 mm | 74.1 mm | 62.6 mm |

The experiment contained three buried spacings (10 m, 15 m and 20 m, recorded as S1, S2, and S3, respectively) of subsurface drainage pipes and six fertilization treatments: 100% organic fertilizer nitrogen (100%OF), 75% organic combined with 25% inorganic fertilizer nitrogen (75%OF), 50% organic combined with 50% inorganic fertilizer nitrogen (50%OF), 25% organic combined with 75% inorganic fertilizer nitrogen (25%OF), 100% inorganic fertilizer nitrogen (0%OF), and no-nitrogen fertilizer (CK). For each treatment, the application of pure nutrient (N, P, or K) was controlled as 120 kg ha$^{-1}$ of N, 85 kg ha$^{-1}$ of $P_2O_5$, and 160 kg ha$^{-1}$ of $K_2O$. Urea, triple superphosphate, and potassium chloride were used as inorganic fertilizers. The organic fertilizers were produced by Nanjing Mingzhu Fertilizer Co., Ltd., Nanjing, China, with N, $P_2O_5$, and $K_2O$ contents of 2.3%, 1.2%, and 1.3%, respectively. For organic fertilizer application treatments, the amounts of P and K brought in by the organic fertilizer were deducted, and the deficient amounts were supplemented by inorganic fertilizer. In order to keep the experimental conditions consistent, the organic and the inorganic fertilizer were applied at the same time. The organic fertilizers were mixed evenly with the surface soil before sowing, and the inorganic fertilizer was applied 3 cm from the seeds after sowing. In summary, this experiment contained a total of 3 (spacing) × 6 (fertilization) = 18 treatments; each treatment was repeated 3 times.

The experiment field was divided into blocks. Each replicate occupied one block. There was a total of 54 blocks in this study. Each block was buried with 5 drainage pipes, with a length of 60 m and a gradient decrease of 0.1%. The area of each block was 120 m × 60 m. A 20 m × 20 m area for harvest crops was installed in the center of each block. The material of the pipe was PVC. The pipes were wrapped with non-woven fabric and laid out with a density of 70 g m$^{-2}$. Except for the different buried spacings of the drainage pipes, other field management measures such as pest control, weeding, etc., were strictly consistent for all the treatments.

*2.3. Sampling and Measurement*

After fertilization and sowing on 8 June, $N_2O$ gas samples were collected three times for three consecutive days. The interval between the fourth collection and the first collection was 15 days, and thereafter the gas samples were collected every 15 days. Samples were collected 11 times in total during the whole growth period. The self-made PVC collection box was used for gas sampling, which included the body and the base. The body was cube-shaped and 40 cm in length, and contained a battery-driven fan to ensure that the gas was mixed uniformly. The base could be inserted into the soil, and the top surface of the base was equipped with a water seal groove. When using the collection box, water was injected into the groove which then covered the body at the base to provide a sealing effect on the gas. There was a gas collection hole on each body, and a 60 mm needle tube was used to collect gas through the collection hole. Each collection time occupied a duration of 5 min, collecting 5 gas samples which were used to calculate the emission flux. A meteorological chromatograph was used to determine $N_2O$ concentration (Agilent 7890D, San Diego, CA, USA).

The $N_2O$ emission flux is calculated as [12]:

$$F = \rho \frac{V}{A} \frac{\triangle c}{\triangle t} \frac{P}{P^0} \frac{273}{273 + T} \times 600$$

where $F$ is the $N_2O$ emission flux, µg m$^{-2}$ h$^{-1}$; $\rho$ is the $N_2O$ density in the standard state, µg m$^{-3}$; $V$ is the volume of collection box, cm$^3$; $A$ is the sampling area, cm$^2$; $P$ is the air pressure inside the collection box, Pa; $P^0$ is the standard atmospheric pressure, $1.01 \times 10^5$ Pa. $\triangle c / \triangle t$ is the concentration variation of $N_2O$ in the collection box per unit time, $10^{-9}$ min$^{-1}$; $T$ is the mean temperature inside the collection box during measurement (°C).

Accumulated $N_2O$ emission is calculated using [1,12]:

$$M = \sum (F_{N+1} + F_N) \times 0.5 \times (t_{N+1} - t_N) \times 24 \times 10^{-2}$$

where $M$ is the accumulated $N_2O$ emission, kg ha$^{-1}$; $N$ is the $N$th sampling; $t$ is the days between the two adjacent measurements. The interpolation method is used to calculate the $N_2O$ emission flux for the unobserved dates between two adjacent measurements. The measured values and the calculated values are added together in order to obtain the accumulated $N_2O$ emission.

The $N_2O$ emission factor is calculated according to the following formula:

$$d = 100(M_N - M_0)/N$$

where d is the $N_2O$ emission factor, $M_N$ is the accumulated $N_2O$ emission with nitrogen application, kg ha$^{-1}$; $M_0$ is the accumulated $N_2O$ emission without nitrogen application, kg ha$^{-1}$; $N$ is the nitrogen application amount, kg ha$^{-1}$.

During the blooming period of sunflowers (75 days after fertilization), the five-point method was used to collect soil samples in the original point where the gas samples were collected. The soil salinity, total carbon, total nitrogen, and organic carbon were observed. The total carbon and organic carbon were determined using the Vario Macro Cube elemental analyzer (Element Company, Frankfurt, Germany). The total nitrogen amount was

measured using the Kjeldahl method [18]. The determining process of salt content was as follows: soil samples were firstly naturally dried, then ground, sieved, extracted (water–soil ratio = 5:1), filtered, and were finally measured the electrical conductivity using DDSJ-308F measurer (produced by Shanghai Leici Instrument Co., Ltd, Shanghai, China). The soil salt content was converted using the electrical conductivity values based on the regression formula obtained by the earlier basic samples from the experimental area. The conversion formula was as follows:

$$S = 4.634EC - 0.626$$

where $S$ is the total salt content, g kg$^{-1}$; $EC$ is the conductivity value, ms cm$^{-1}$.

At harvest stage of sunflowers, the weight of their flower discs and the weight of 100 seeds were measured, and then converted into the yield (t ha$^{-1}$) for analysis. The sunflower plants were killed at a high temperature of 105 °C for half an hour and dried at 70 °C to a constant weight. After grinding, the plant samples were digested using concentrated $H_2SO_4$, and the plant nitrogen levels (kg ha$^{-1}$) were measured using the Kjeldahl method [19].

### 2.4. Statistical Analysis

The data from the soil and plant indicators were submitted to the SPSS17.0 software for calculating the significant differences (according to Duncan's multiple ranging test).

## 3. Results and Analysis

### 3.1. $N_2O$ Emission Flux and Accumulative $N_2O$ Emission

The $N_2O$ emission flux reached a short-term peak after fertilization, followed by a decrease, and finally reached stabilization (Figure 1), with a relatively consistent trend for all the treatments (Figure 1a–c). Under the same fertilization treatment, the $N_2O$ emission flux was in a higher level under S3; this was more evident from 0 to 2 days after fertilization. The maximum $N_2O$ emission flux occurred on the first day after fertilization that was detected in the 0%OF treatment, reaching as high as 1.76 mg m$^{-2}$ h$^{-1}$.

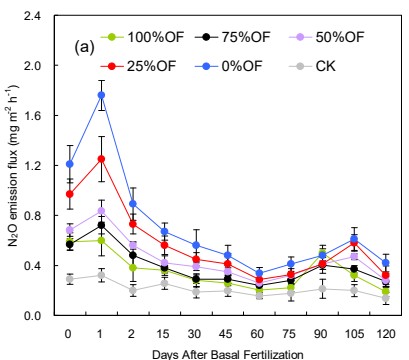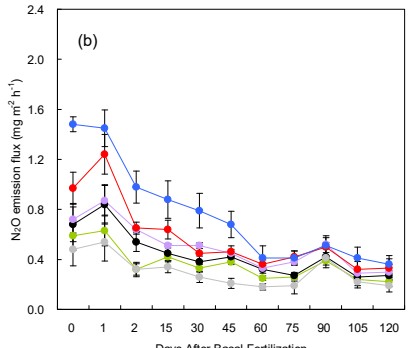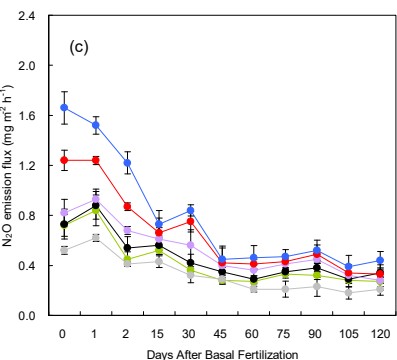

**Figure 1.** Variation in $N_2O$ emission flux with days after basal fertilization under 10 (**a**), 15 (**b**), and 20 m (**c**) buried spacings of drainage pipes (100%OF, 75%OF, 50%OF, 25%OF, and 0%OF indicate that 100%, 75%, 50%, 25%, and 0% nitrogen came from organic fertilizer. CK refers to non-fertilization treatment. All data are mean ± SD).

After fertilization, the $N_2O$ emission flux from each fertilization treatment was significantly ($p < 0.05$) higher than that from CK, indicating that fertilizer application promoted $N_2O$ emissions. Under the same spacing, the increased application ratio of organic fertilizer lowered the $N_2O$ emissions, and this regularity was found under all S1–S3 spacings. At 120 days after fertilization, the $N_2O$ emission flux under the treatments reached a relatively low level. The highest $N_2O$ emission fluxes under S1, S2, and S3 all appeared as a result of 0%OF treatment, with values of 0.42, 0.36, and 0.51 mg m$^{-2}$ h$^{-1}$, respectively.

The cumulative $N_2O$ emissions can reflect the impact of different fertilization treatments on $N_2O$ emissions more clearly (Figure 2). The cumulative $N_2O$ emissions for the

treatments showed a linear upward trend. At 120 days after fertilization, the cumulative $N_2O$ emissions from the fertilization treatments under S1, S2, and S3 were 8.9–15.8, 9.5–17.5, and 10.1–17.6 kg ha$^{-1}$, respectively, significantly higher than that from CK (5.8, 7.6, and 8.0 kg ha$^{-1}$). The highest cumulative $N_2O$ emission occurred in the 0%OF treatment at S3, reaching a value of 17.6 kg ha$^{-1}$.

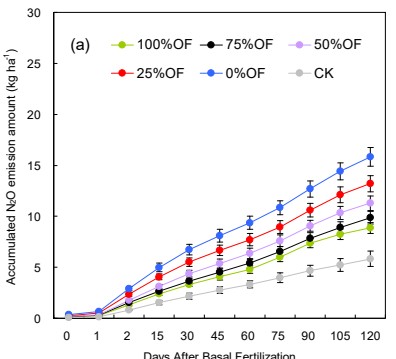 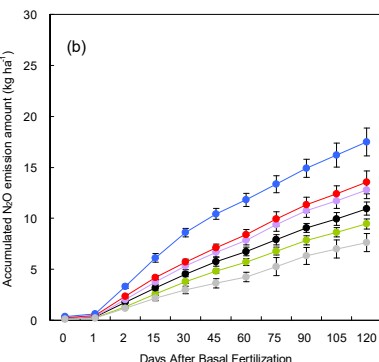 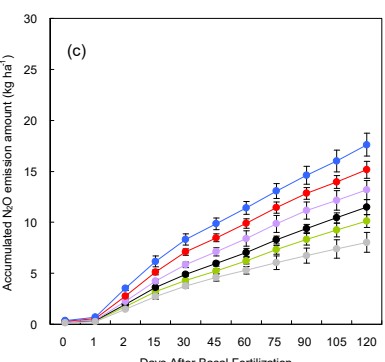

**Figure 2.** The accumulated $N_2O$ emission with days after basal fertilization under 10 (**a**), 15 (**b**), and 20 m (**c**) buried spacings of drainage pipes (100%OF, 75%OF, 50%OF, 25%OF, and 0%OF indicate that 100%, 75%, 50%, 25%, and 0% nitrogen came from organic fertilizer. CK refers to non-fertilization treatment. All data are mean ± SD).

The effect of fertilization on the cumulative $N_2O$ emission was similar to the $N_2O$ emission flux, showing that the higher proportion of organic fertilizer reduced the cumulative $N_2O$ emission. On the contrary, the higher application proportion of inorganic fertilizer promoted the cumulative $N_2O$ emission.

The greater usage of inorganic fertilizer resulted in the greater $N_2O$ emission coefficient under subsurface drainage (Figure 3). The highest $N_2O$ emission coefficient was more than three times higher compared with the lowest. The $N_2O$ emission coefficient under the 0%OF treatment was the highest, ranging from 8.0% to 8.4%; followed by 25%OF, recording as 4.9–6.2%; the $N_2O$ emission coefficient under 100%OF was in the lowest level, with values ranging from 1.5% to 2.5%.

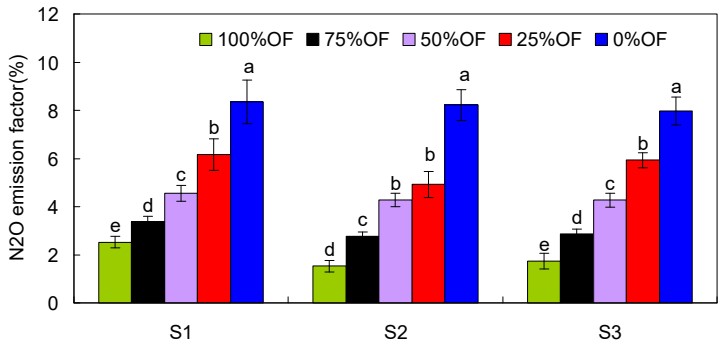

**Figure 3.** The $N_2O$ emission factor under combined application of organic and inorganic fertilizers (100%OF, 75%OF, 50%OF, 25%OF, and 0%OF indicate that 100%, 75%, 50%, 25%, and 0% nitrogen came from organic fertilizer. CK refers to non-fertilization treatment. All data are mean ± SD. Different values (a, b, c, d, e) mean significant differences at a level of 0.05 according to Duncan's multiple range test.).

### 3.2. Soil Chemical Indicators

When under the same fertilization treatment, a smaller distance between drainage pipes was beneficial for removing salt from the topsoil in the view of average salinity (Figure 4a). The salt content in the topsoil treated with S1 ranged from 2.25 to 2.98 g kg$^{-1}$, which was generally lower than that with S2 or S3. The fertilization promoted the increase

in salt content in the topsoil. The treatment with lower proportion of organic fertilizer increased the soil salt content more obviously. Compared with CK, the soil salinity under the 0%OF treatment was increased by an average of 32.5%, whereas it increased by only 8.9% under the 100%OF treatment.

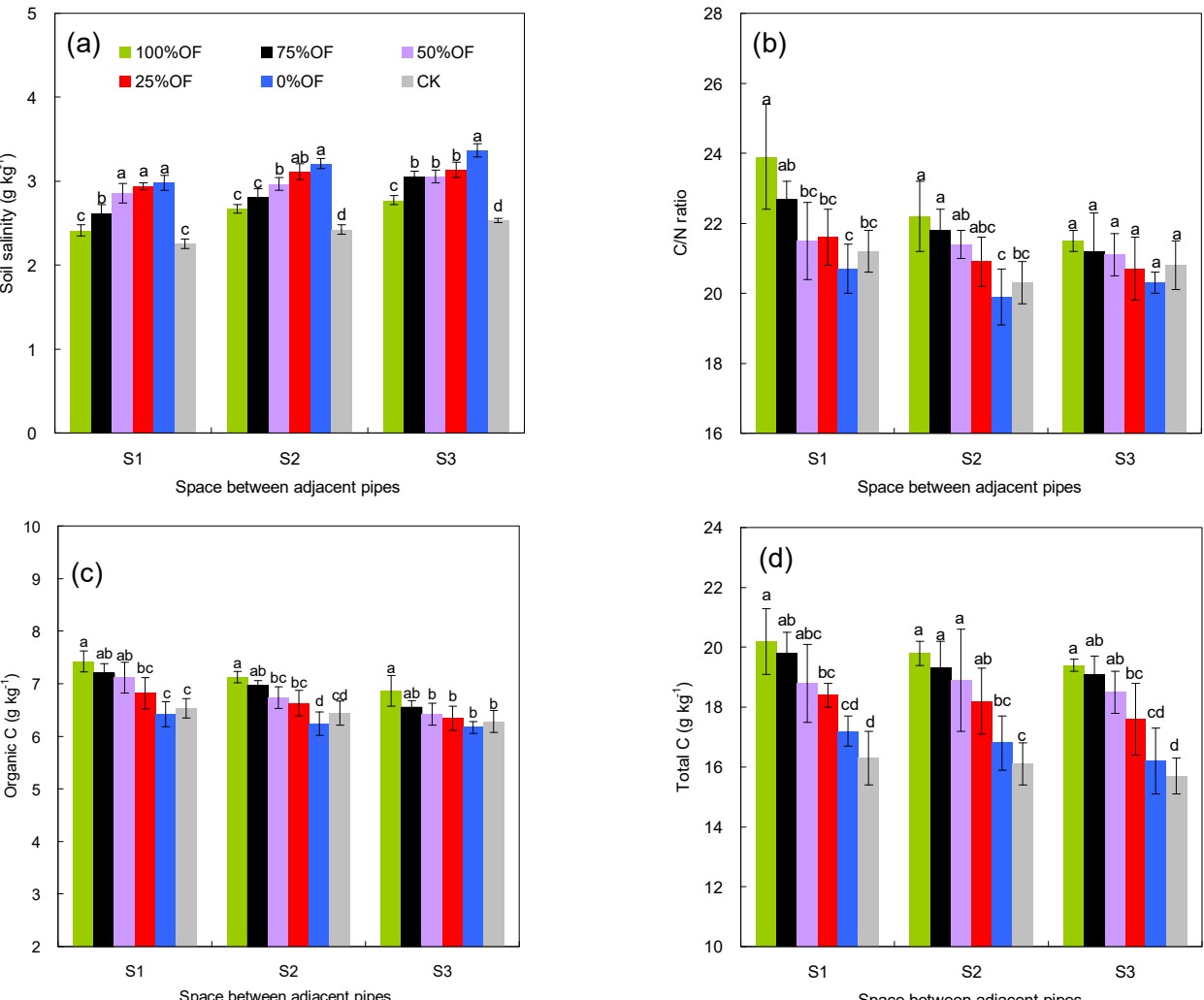

**Figure 4.** Effects of combined application of organic and inorganic fertilizers on soil salinity (**a**), C/N ratio (**b**), organic C (**c**), and total C (**d**) under 10 (S1), 15 (S2), and 20 m (S3) buried spacings of drainage pipes (100%OF, 75%OF, 50%OF, 25%OF, and 0%OF indicate that 100%, 75%, 50%, 25%, and 0% nitrogen came from organic fertilizer. CK refers to non-fertilization treatment. All data are mean ± SD. Different values (a, b, c, d) mean significant difference at 0.05 level according to Duncan's multiple range test).

The application of organic fertilizer increased the soil C/N ratio, which was found at S1, S2, and S3 (Figure 4b), but applying 50% or less than 50% of organic fertilizer did not significantly increase the C/N ratio compared with CK. Overall, under the same fertilization treatment, the smaller spacing between pipes resulted in a higher soil C/N ratio. The highest soil C/N ratio reached 23.9 in the 100%OF treatment under S1.

The response of organic carbon and total carbon to different treatments was similar to that of the C/N ratio (Figure 4c,d). In general, the differences in organic carbon or total carbon among 100%OF, 75%OF, and 50%OF treatments were not significant ($p > 0.05$). However, there was a significant ($p < 0.05$) difference in organic carbon or total carbon between 100%OF and 25%OF. Compared with CK, the amount of organic carbon or total carbon under the 0%OF treatment did not change significantly ($p < 0.05$). The organic

fertilizer application treatments (25–100%OF) increased the organic carbon by 1.0–13.6% and the total carbon by 3.2–23.9%. Moreover, under the same fertilization treatment, the soil organic carbon or total carbon with S1 was slightly higher.

### 3.3. Crop Yield and Nitrogen Uptake

For all three spacings, the crop yield firstly increased and then decreased as the organic fertilizer ratio increased (Table 2). The crop yield under 25%OF was at the highest level, and reached 3.41, 3.82, and 3.27 t ha$^{-1}$ under S1, S2, and S3, respectively. The yield under CK was 2.30–2.71 t ha$^{-1}$, which was generally at the lowest level. Compared with CK, the fertilization treatments increased crop yield by 8.9–48.3%. Under the same fertilization treatment, the yield of sunflowers with S2 was overall higher. The highest yield was observed with S2 under 25%OF fertilization treatment.

**Table 2.** Crop yield and plant nitrogen absorption.

| Treatment | S1 | | S2 | | S3 | |
|---|---|---|---|---|---|---|
| | Crop Yield (t ha$^{-1}$) | Nitrogen Absorption (kg ha$^{-1}$) | Crop Yield (t ha$^{-1}$) | Nitrogen Absorption (kg ha$^{-1}$) | Crop Yield (t ha$^{-1}$) | Nitrogen Absorption (kg ha$^{-1}$) |
| 100%OF | 2.74 ± 0.11 d | 134.9 ± 15.1 a | 2.95 ± 0.11 cd | 142.3 ± 15.4 a | 2.65 ± 0.11 bc | 144.6 ± 14.1 a |
| 75%OF | 2.93 ± 0.14 cd | 139.1 ± 14.0 a | 3.13 ± 0.12 c | 135.2 ± 10.3 a | 2.85 ± 0.18 b | 138.1 ± 9.5 a |
| 50%OF | 3.15 ± 0.11 bc | 137.9 ± 9.3 a | 3.43 ± 0.16 b | 132.7 ± 8.9 a | 3.11 ± 0.10 a | 142.3 ± 3.9 a |
| 25%OF | 3.41 ± 0.19 a | 137.2 ± 11.1 a | 3.82 ± 0.17 a | 132.5 ± 9.9 a | 3.27 ± 0.12 a | 138.3 ± 6.2 a |
| 0%OF | 3.22 ± 0.12 ab | 127.7 ± 15.1 ab | 3.56 ± 0.14 ab | 125.9 ± 3.6 ab | 3.18 ± 0.13 a | 133.4 ± 6.6 a |
| CK | 2.30 ± 0.09 e | 106.6 ± 5.0 b | 2.71 ± 0.15 d | 108.8 ± 8.1 b | 2.43 ± 0.12 c | 113.5 ± 11.4 b |

Note: 100%OF, 75%OF, 50%OF, 25%OF, and 0%OF indicate that 100%, 75%, 50%, 25%, and 0% of applied nitrogen comes from organic fertilizer. CK refers to non-fertilization treatment. All data are mean ± SD. Different letters (a, b, c, d, e) indicate significant differences at 0.05 level. S1, S2, and S3 represent the three different buried spacings of 10, 15, and 20 m for subsurface drainage pipes.

The fertilizer application obviously increased the nitrogen uptake of sunflower plants, with an increase of 18.1–47.2%, 8.6–40.5%, and 8.8–34.5% under S1, S2, and S3, respectively, compared with CK. The 25%OF treatment achieved the highest amount of plant nitrogen, but the difference was not significant ($p > 0.05$) compared with 0%OF or 50%OF (excluding under S2). When the fertilization strategy was the same, the spacing between pipes also has an impact on plant nitrogen absorption, S2 has a higher amount of plant nitrogen (157.0 kg ha$^{-1}$), which was higher compared with S1 (141.9 kg ha$^{-1}$) and S3 (140.6 kg ha$^{-1}$).

### 3.4. Correlation between N$_2$O Emissions and Possible Influencing Factors

A significant positive correlation was found between N$_2$O emission flux and soil salinity, with a correlation coefficient of 0.96 (Figure 5a). Overall, the N$_2$O emission flux was negatively correlated with the soil C/N ratio, total carbon or organic carbon (Figure 4b–d). Furthermore, a positive correlation was detected between the N$_2$O emission flux and the crop yield or the plant nitrogen uptake, with correlation coefficients of 0.76 or 0.50, respectively.

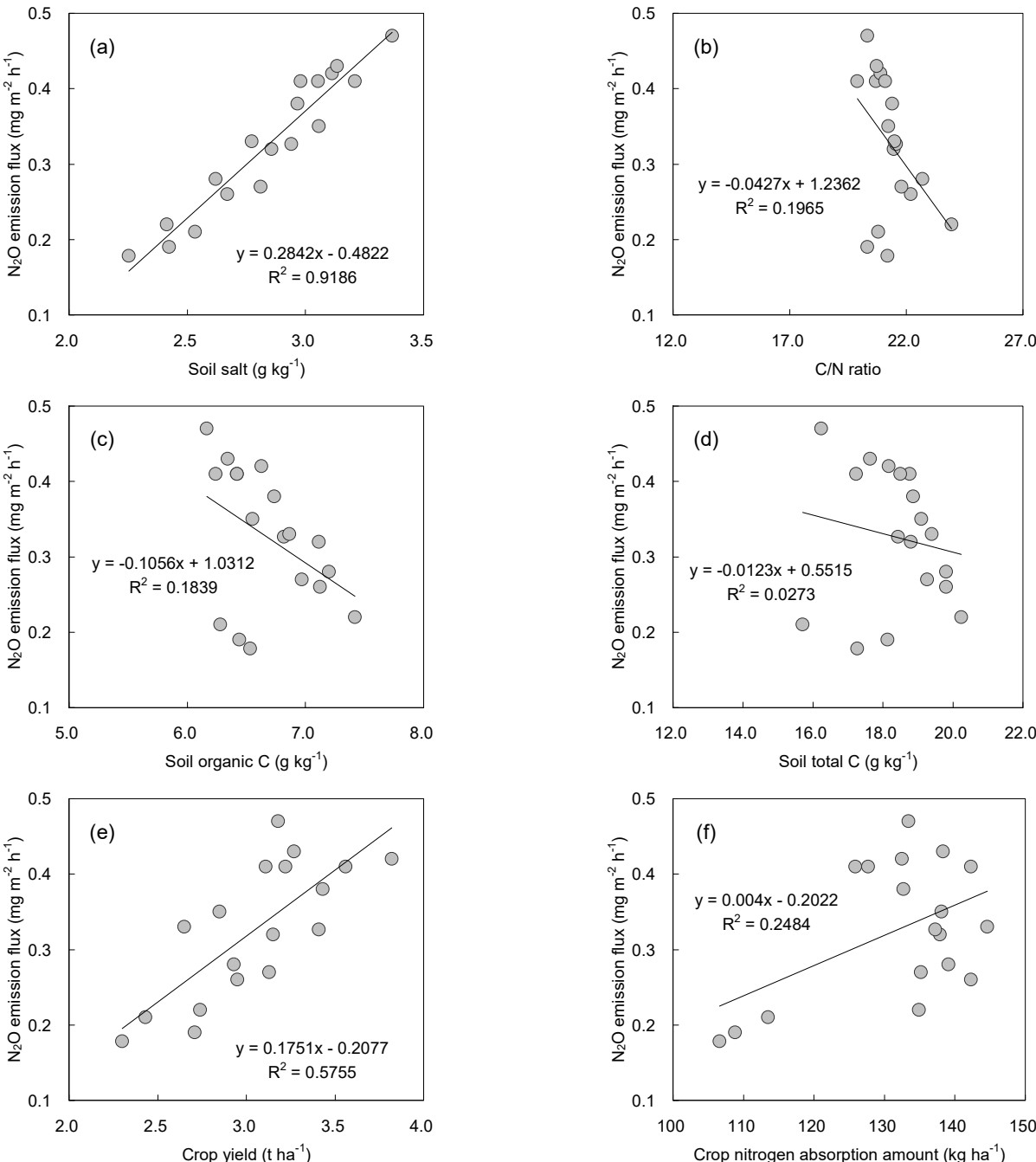

**Figure 5.** Relationship between N$_2$O emission flux and possible influencing factors (Figure a–f correspond to the relationship between N$_2$O emission flux and soil salinity (**a**), C/N ratio (**b**), soil organic C (**c**), soil total C (**d**), crop yield (**e**), and crop nitrogen absorption amount (**f**). The grey circles and black line display the quadratic linear relationship).

## 4. Discussion

The coastal saline soil has higher salt content, lower porosity and deficient available oxygen, creating a different state compared with the inland cultivated soil. A study [20] showed that the gaseous loss of nitrogen in shallow coastal saline soil was driven by the anaerobic ammoxidation process rather than the denitrification process; this confirmed that the dominant genus of Anammox resulted in the gaseous nitrogen loss was *Candidatus Scanlindua*, indicating that the application of organic fertilizer on coastal saline soil was an effective measure to increase the number of beneficial microorganisms and improve

microbial community for reducing the nitrogen loss driven by microorganisms. The subsurface drainage was beneficial for lowering the groundwater level, improving the soil permeability, and creating favorable conditions for crop roots to absorb and fix the nitrogen [21,22].

Our experiment observed the $N_2O$ emissions during the 120 days of crop growth period, mainly due to the fact that the frequent activities such as irrigation, drainage, and fertilization in this period have the more significant impacts on the soil $N_2O$ emissions. This study found that the higher application ratio of organic fertilizer decreased the $N_2O$ emissions, which was similar to Wang's [23] conclusion. Wang's research showed that under the application of 50% chemical fertilizer combined with 50% organic fertilizer as well as 30% chemical fertilizer combined with 70% organic fertilizer, the annual average soil $N_2O$ emissions (15.8 and 14.4 kg N ha$^{-1}$) decreased by 21% and 28%, respectively. Yan's [24] research showed that the 30% chicken manure organic fertilizer could not only reduce $N_2O$ emissions and $NH_3$ volatilization in the protected field, but also ensure the stable output of vegetables. Liu's [25] experiment demonstrated that long-term application of chemical fertilizers (N, NK, NPK) significantly increased cumulative soil $N_2O$ emissions. Shao [2] noted that when 50–100% organic fertilizers were used instead of chemical fertilizer, $N_2O$ emissions from vegetable cultivated soil were reduced by 26.3–40.2%, respectively. However, Chen [26] found that the application of pig manure fertilizer significantly increased the soil $N_2O$ emission in northern China, but chicken manure has no significant impact on the $N_2O$ emission. The differences in the above studies were mainly due to the differences in the composition and structure of organic fertilizer.

In addition, this study also found that the smaller distance between drainage pipes lowered the $N_2O$ emissions to different degrees. This was because the smaller distance created the smoother drainage, and the soluble nitrogen in the topsoil would have a greater probability to be discharged along with the drainage water [10]. The main components of soluble nitrogen were the nitrate nitrogen and the ammonium nitrogen [27,28], which were important substrates for $N_2O$ emissions. The greater loss amount of soluble nitrogen resulted in the lower concentration of substrate for $N_2O$ emissions, thus limiting the $N_2O$ emissions.

The impact of the factors such as temperature and humidity on the soil $N_2O$ emissions has been widely studied; therefore, this study focused on the characteristics of saline soil and explored the impact of soil chemical indicators such as soil salinity on the $N_2O$ emissions. This study found a significant positive correlation between the $N_2O$ emission amount and the total salt content in the topsoil, which was in line with Wen's [29] results. Wen's research on four different saline soils found that the $N_2O$ emission rate increased as the soil salinity (conductivity) increased. Reddy [30] also reached the same conclusion. The reason might be that in the saline soil, the $N_2O$ reductase activity was limited by soil salinity and the $N_2O$ was easier to accumulate and emit. In addition, this study found that the increased proportion of organic fertilizer increased the soil total carbon and C/N ratio, which was because the organic matter in organic fertilizer contained a lot of carbon elements, and the application of organic fertilizer increased the input of soil C. The latest global scale meta-analysis [31] showed that compared with chemical fertilizer, the application of organic fertilizer could increase the storage of soil organic C by 7.41 Mg ha$^{-1}$ in the farmland. Meanwhile, our study found that under treatments with a high soil C or C/N ratio, the $N_2O$ emissions were lowered, which confirmed many previous research findings [32,33]. When the C or C/N ratio was at a high level, soil microbial activity would be more vigorous, consuming greater amount of substrates such as mineral N, thus reducing the emission of $N_2O$ [34]. The earlier research proved that the availability and dynamics of carbon substrate have a greater impact on $N_2O$ emissions than the changes in soil moisture (dry and wet cycling) [35,36].

The crop yields treated with 25%OF and 0%OF were relatively higher in this study, indicating that inorganic fertilizer was the main promoting factor for the increase in crop yield. Previous studies have shown that the effect of combined application of organic and

inorganic fertilizers was better than that of single application of inorganic fertilizer [37]. The main reason was that the combined application could make up for the difference in the release rates of different nutrients, maintaining the supply of effective nutrients and increasing the yield [38]. Our result was consistent with the previous research findings [39,40]. However, the crop yield decreased when the proportion of organic fertilizer exceeded 50%, this might because most of the nutrients in organic fertilizer were not directly absorbable, which could only be used after microbial decomposition and mineralization. The nutrient release lagged behind the needs of the plant, thus affecting the yield formation. Under the same fertilization treatment, both S1 and S3 treatments obtained lower crop yields than S2. This might be due to the greater spacing between drainage pipes, which led to relatively poor drainage, retaining more salts in the topsoil, which limited the crop growth and thus restrained the yield formation. However, if the distance between pipes was too small, the drainage would be enhanced; it was easier to cause the loss of available nitrogen from the top soil layer, which limited the plant nitrogen absorption and thus reduced the crop yield.

This study found a positive correlation between the crop yield/plant nitrogen uptake and the $N_2O$ emission flux (Figure 5). The higher crop yield or the greater plant nitrogen absorption suggested that there was a higher content of available nitrogen in soil. The soil available nitrogen (nitrite nitrogen, ammonium nitrogen, nitrate nitrogen) was the important substrates for $N_2O$ emissions [2,41], which could promote the soil nitrification and denitrification processes, thereby increasing $N_2O$ emissions. This indicated that when the nitrogen behavior in the surface soil was more active, the plant would absorb greater amount of nitrogen absorption, and the soil would emit a greater amount of $N_2O$. An early study [42] found that crop growth might have a dual effect on the soil $N_2O$ emissions, namely both promoting and inhibiting soil $N_2O$ emissions. On one hand, crop growth would compete with soil microorganisms to absorb nitrogen in the soil, so that a lack of mineral nitrogen for nitrification or denitrification would reduce the soil $N_2O$ emissions; on the other hand, the growth of plant roots could consume the oxygen from the soil, change the soil structure, and secrete the organic matter, thereby promoting the progress of microbial denitrification and stimulating the production of $N_2O$.

Subsurface drainage is an important means of desalination in the coastal saline soil, and the fertilizer application can compensate for the nutrient loss caused by subsurface drainage in the topsoil [11]. Our research has proven that subsurface drainage and the organic–inorganic fertilization have an effective impact on the improvement of saline soil and crop yield. Moreover, we emphasize that the reasonable layout of subsurface drainage pipes and the scientific application of organic fertilizer can effectively reduce the $N_2O$ emissions from the soil. One detriment to this study is that we have not monitored the nitrogen indicators in the drainage water. A smaller distance between pipes resulted in fewer soil $N_2O$ emissions; however, this may also cause a greater risk of soil nitrogen loss and a higher cost. Therefore, it is important to comprehensively consider the possible factors and optimize the layout of hidden pipes through a multi-scheme decision model. Another deficit of this study is the low frequency during the early collection. Studies have shown that higher $N_2O$ flow rates occur within a few days (more than 3 days) after the application of fertilizer.

Future research should note that in the actual production, when the coastal agricultural area is close to the livestock and poultry breeding area, the source of organic fertilizer may mostly be the animal manure fertilizer; when close to the crop production area, where the straw yield is huge, the source of organic fertilizer is more likely to be the plant straw organic fertilizer. In the future, it is worth studying this further in order to distinguish the differences in $N_2O$ emissions from coastal saline soil caused by different sources of organic fertilizer and find out the scientific mechanism behind the differences.

## 5. Conclusions

Our overall results showed that the cumulative $N_2O$ emissions from the treatments under S1, S2, and S3 throughout the entire growth period were 8.9–15.8, 9.5–17.5, and

10.1–17.6 kg ha$^{-1}$, respectively. A smaller spacing between adjacent drainage pipes or a higher application proportion of organic fertilizer reduced the accumulative N$_2$O emissions. The increased application of organic fertilizer decreased the soil salinity, whereas it increased the C/N ratio and total carbon content. The fertilization treatments significantly increased the nitrogen uptake of sunflower plants, with increase ranges of 18.1–47.2%, 8.6–40.5%, and 8.8–34.5% under S1, S2, and S3, respectively, compared with CK. The highest yield of sunflowers was achieved under S2 combined with 25%OF, reaching 3.82 t ha$^{-1}$. Correlation analysis showed that N$_2$O emission flux was positively correlated with the soil salinity, crop yield, and crop nitrogen uptake, whereas it was negatively correlated with the total carbon, C/N ratio, and organic carbon content. We highlighted the important role of subsurface drainage, organic fertilizer application, and sunflower planting for the sustainable development of coastal agriculture. We concluded that using 25% organic fertilizer instead of inorganic fertilizer was beneficial for reducing N$_2$O emissions while maintaining the crop yield. However, the spacing of drainage pipes should be determined based on a comprehensive consideration of N$_2$O emissions, nitrogen loss, and engineering costs.

**Author Contributions:** Conceptualization Y.Z.; methodology, Y.X.; software, L.Z.; validation, Q.Z. and Y.X.; formal analysis, J.C.; investigation, M.H.; resources, Q.Z.; data curation, J.C.; writing—original draft preparation, Y.Z.; writing—review and editing, Y.Z.; visualization, M.H.; supervision, J.C.; project administration, M.H.; funding acquisition, L.Z. All authors have read and agreed to the published version of the manuscript.

**Funding:** This work was funded by the cross-fusion project "Green Civil Engineering and Water Conservancy" (000-71202103D) of Fujian Agricultural and Forest University; Young and Middle-aged Project of Education Department of Fujian Province (JAT210784).

**Data Availability Statement:** Not applicable.

**Acknowledgments:** We appreciate three reviewers in particular for their thoughtful and helpful comments on the manuscript for further improvement.

**Conflicts of Interest:** The authors declare no conflict of interest.

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
