# Peer review of "N2O Emissions from Saline Soils in Response to Organic–Inorganic Fertilizer Application under Subsurface Drainage"

_water, doi:10.3390/w15163002_

Round 1
Reviewer 1 Report
The paper entitled "N2O emission from saline soils in response to organic-inorganic fertilizer application under subsurface drainage" have a great importance and nicely designed piece of work. There are few concerns that should be taken care
Abstract- Probably there is typographical error in the first line of the abstract "Organic fertilizer application sand subsurface drainage are two...." it should be "Organic fertilizer applications and subsurface drainage are two...".
Page 1 line No. 12-13. Re-phrase the sentence "but while N2O emission from saline soil under these two measures is seldom evaluated".
Is the results are from one year data only?
Line No. 80-86. Re-phrase the paragraph, as it is tough to understand meaning.
Almost all the sections of the manuscript need careful reading by the authors and improvements.
The heading "2.4. Statistical analysis" needs to be re-written.
The reference is not according to journal guidelines.
The English is not up to the journal standards. Critical reading and correction needed
Author Response
- The paper entitled "N2O emission from saline soils in response to organic-inorganic fertilizer application under subsurface drainage" have a great importance and nicely designed piece of work. There are few concerns that should be taken care.
Thank you for your affirmation and valuable advice on our article.
- Abstract- Probably there is typographical error in the first line of the abstract "Organic fertilizer application sand subsurface drainage are two...." it should be "Organic fertilizer applications and subsurface drainage are two...".
Yes, thank you, it is a fault, we have revised it.
- Page 1 line No. 12-13. Re-phrase the sentence "but while N2O emission from saline soil under these two measures is seldom evaluated".
Thank you for your suggestion. We have deleted the “but”, and the new version is :while N2O emission from saline soil under combination of these two measures is seldom evaluated.
- Is the results are from one year data only?
Yes, in this year, the soil was mixed evenly with a rotary tiller, and the experimental conditions were controlled with high precision.
- Line No. 80-86. Re-phrase the paragraph, as it is tough to understand meaning.
Thank you for your reminding. We have revised, and now the new version is : In summary, there are some differences among the previous studies on how the fertilizers affect N2O emissions, and the mechanisms are needed to be further studied. Although many studies have evaluated the impact of the organic fertilizer replacing chemical fertilizer on N2O emissions, most of them were conducted in the inland field. For coastal land, a relatively new and special agricultural land, whether there are differences in the regularity of N2O emission, how the N2O emissions are affected by the saline soil agronomic measures such as salt-absorbing crop planting, subsurface drainage, as well as the organic fertilizer application), are not clear.
- Almost all the sections of the manuscript need careful reading by the authors and improvements.
Thank you for your reminding. We have carefully examined the MS, and made the corresponding revisions, as you can see from the revised paper.
- The heading "2.4. Statistical analysis" needs to be re-written.
Thank you for your reminding. Duncan's multiple ranging test is a choice of the software. We have rewritten the 2.4 as: The data were submitted to the SPSS17.0 software for calculating the significant differences (according to Duncan's multiple ranging test).
- The reference is not according to journal guidelines.
Thank you for your suggestion. After all the references have been modified, we will use endnote software to rearrange them all at once.

Reviewer 2 Report
This study includes results of only one year of a field trial that allows to identify the N2O emissions from a saline soil, with drainage pipes installed with three different spacing, and producing a sunflower crop fertilized with different combinations of organic and inorganic fertilizers.
Considering the research areas of “Water”, probably the paper may not be very well within the scope of this journal. This does not mean that the authors have not presented an interesting work
As the authors themselves recognize (line 44), the coastal soils pose a series of constraints to agricultural production. Therefore, I think that in the introduction the authors should clarify the interest in using this type of soil (with primary salinity) for “conventional” agriculture and, thus, better reinforce the interest of the study carried out.
Sunflower is a crop with moderate salt-tolerance. It is not a tolerant crop. Thus, in the introduction, the authors should better justify the option of using this crop in the experiment.
In my opinion, in Materials and methods some information should be added, such as: municipal waters characteristics (e.g. nitrogen content and EC), cultivar of sunflower (differ in their tolerance to salinity) and irrigation system used, area of each plot and harvested crop area, cultural practices adopted to fertilizer’s soil incorporation and crop installation. All the inorganic nitrogen fertilizer was applied at sowing?
The soil already has considerable levels of N and K. Have these levels been taken into account when fertilizing the crop?
From my point of view, the results are adequately presented but its statistical analysis must be clarified (or even improved).
In Materials and methods the statistical analysis must be more informative. For instance, the authors say that “Data were submitted to SPSS17.0 software for significance analysis according to Duncan's multiple ranging test.” (line 188). We don’t see that in what concern to the N2O emissions results. For these results, only the standard deviation was considered (for the accumulated N2O emissions not even that). Why? Before Post-Hoc test by Duncan, did you verify significant differences between means? And before that did you checked if the data for each individual group followed a normal distribution?
The authors claim (line 228) that “When under the same fertilization treatment, a smaller distance between drainage pipes was beneficial for removing salt from the topsoil (Figure 3-a)”. I confess that I look at the figure and for me such conclusion is not clear. Did you analyze, statistically, the differences between pipes distances? Or only between fertilization treatments? The same comment can be made regarding other results.
A total of 11 times of samples were collected during the whole reproductive period. In my opinion, this is a scarce number of observations, namely in the period following the incorporation of fertilizers (only for 3 days). There are many reports of observations of higher flows occurring several days (more then 3) after the fertilizers incorporation.
The authors associated higher crop yields with higher N2O emissions. However, I think it would be important to calculate the N2O emission factor in the different treatments.
The observation that “… 25% organic fertilizer instead of inorganic fertilizer was beneficial for reducing N2O emissions while maintaining the crop yield.”, it is a good and important conclusion. However, at the end of a study like this, the authors should be able to clarify (discuss) more about the use of drainage pipes, as a way to reduce N2O emissions. It is mentioned (line 319) that “this study also found that a smaller distance between drainage pipes lowered the N2O emissions…”, as a result of “… the greater probability of soluble nitrogen in the top-soil would be discharged along with the drainage water.”. If it happened in this experiment (drainage and nitrogen leaching were not measured), is this a viable consequence? Will be a solution to solve a problem by creating another?
Additional comments
. line 11 – “Organic fertilizer application sand subsurface drainage are two important…”. Review the text.
. line 17 “…50% organic fertilizer (50% OF), 25% organic fertilizer (25% OF) …” Review the text.
. line 31 – “Keywords: N2O, organic fertilizer, subsurface drainage, saline soil, sunflower” With the exception of the word “sunflower”, all the others appear in the title. I suggest its replacement by other keywords
. line 207 – “Figure 1. Variation of N2O emission flux with days after transplanting …”. Transplanting?
Author Response
- This study includes results of only one year of a field trial that allows to identify the N2O emissions from a saline soil, with drainage pipes installed with three different spacing, and producing a sunflower crop fertilized with different combinations of organic and inorganic fertilizers.Considering the research areas of “Water”, probably the paper may not be very well within the scope of this journal. This does not mean that the authors have not presented an interesting work.
Thank you very much for your guidance and affirmation of our content. Your opinion has greatly helped us improve our MS.
- As the authors themselves recognize (line 44), the coastal soils pose a series of constraints to agricultural production. Therefore, I think that in the introduction the authors should clarify the interest in using this type of soil (with primary salinity) for “conventional” agriculture and, thus, better reinforce the interest of the study carried out.
Thank you for your suggestion, indeed, to reinforce the interest of the study is very important. In the revised paper, we noted that why the type of soil is useful for cultivation. As: The coastal soil is formed by marine sediments under the action of ocean tides or high concentrations of groundwater. Therefore, some coastal soils in certain regions have advantages such as high organic matter content and abundant mineral nutrients.
- Sunflower is a crop with moderate salt-tolerance. It is not a tolerant crop. Thus, in the introduction, the authors should better justify the option of using this crop in the experiment.
Thank you very much for your suggestion. We have included your suggestion in the revised manuscript. Indeed, sunflowers are moderately salt tolerant plants, in practice in China, including in project areas, sunflowers are widely planted possibly because they have a certain absorption capacity. We have revised the content of the article as: Sunflower is only a crop with moderate salt-tolerance, while it is widely planted in coastal area for improving saline soil due to its absorption ability.
- In my opinion, in Materials and methods some information should be added, such as: municipal waters characteristics (e.g. nitrogen content and EC), cultivar of sunflower (differ in their tolerance to salinity) and irrigation system used, area of each plot and harvested crop area, cultural practices adopted to fertilizer’s soil incorporation and crop installation. All the inorganic nitrogen fertilizer was applied at sowing?
Thank you very much for your valuable suggestion. Indeed, we have some omissions. Now in the revised version, we have added one by one according to your suggestion,, as follows:
The municipal waters with 0.3 mg L-1 nitrogen content and 0.6 ms cm-1 EC were used for irrigation. The furrow irrigation was employed.
The experiment used sunflower variety “Baikuiza 6” (Helianthus annuus L.) as the plant material.
The area of each block is 120 m×60 m. An area for harvest crop with 20 m×20 m is installed in the center of each block.
In order to keep the experimental conditions consistent, the organic and the inorganic fertilizer were applied at one time. The organic fertilizers were mixed evenly with the surface soil before sowing, and the inorganic fertilizer were applied 3 cm beside the seeds after sowing.
- The soil already has considerable levels of N and K. Have these levels been taken into account when fertilizing the crop?
Thank you very much for your question. We did not consider the nutrient elements in the soil itself, and we still followed the local fertilization habits for the amount.
- From my point of view, the results are adequately presented but its statistical analysis must be clarified (or even improved).In Materials and methods the statistical analysis must be more informative. For instance, the authors say that “Data were submitted to SPSS17.0 software for significance analysis according to Duncan's multiple ranging test.” (line 188). We don’t see that in what concern to the N2O emissions results. For these results, only the standard deviation was considered (for the accumulated N2O emissions not even that). Why? Before Post-Hoc test by Duncan, did you verify significant differences between means? And before that did you checked if the data for each individual group followed a normal distribution?
Thank you for your suggestion. Yes, indeed, we have not expressed clearly. The significant differences are used in the soil indicators such as the salinity and soil C we measured, but not in N2O evaluation. And the data was tested before analysis. Based on your feedback, we have specified the usage scenarios for Duncan as follows: The data of soil or plant indicators were submitted to the SPSS17.0 software for calculating the significant differences (according to Duncan's multiple ranging test). Thank you.
- The authors claim (line 228) that “When under the same fertilization treatment, a smaller distance between drainage pipes was beneficial for removing salt from the topsoil (Figure 3-a)”. I confess that I look at the figure and for me such conclusion is not clear. Did you analyze, statistically, the differences between pipes distances? Or only between fertilization treatments? The same comment can be made regarding other results.
Thank you very much for your guidance. Indeed, our statement can easily lead to ambiguity. Smaller spacing refers to the smaller one of 10, 15 and 20 m in our design of experiment. Under the same fertilization treatment, the regularity of smaller spacing leading to smaller salt content can be observed generally, and the average value can also obtain the conclusion. Therefore, based on your opinion, in order to prevent ambiguity, we have made a supplement after the sentence as: from the view of average salinity among fertilization treatments
- A total of 11 times of samples were collected during the whole reproductive period. In my opinion, this is a scarce number of observations, namely in the period following the incorporation of fertilizers (only for 3 days). There are many reports of observations of higher flows occurring several days (more then 3) after the fertilizers incorporation.
Thank you very much for your suggestion, which has given us a lot of inspiration. Indeed, this is a deficiency of our research. We highly value your opinion, and we added this paragraph to the discussion: In our future research, the frequency of N2O collection will be increased. One drawback of this study is the low frequency during the early collection. Studies have shown that higher N2O flow rates occur within a few days (more than 3 days) after fertilizer applied.
- The authors associated higher crop yields with higher N2O emissions. However, I think it would be important to calculate the N2O emission factor in the different treatments.
Thank you very much for your suggestion. We have calculated the N2O emission factor and added a Figure as Figure 3: The N2O emission factor under combined application of organic and inorganic fertilizers. We also added the calculation method of N2O factor.
- The observation that “… 25% organic fertilizer instead of inorganic fertilizer was beneficial for reducing N2O emissions while maintaining the crop yield.”, it is a good and important conclusion. However, at the end of a study like this, the authors should be able to clarify (discuss) more about the use of drainage pipes, as a way to reduce N2O emissions. It is mentioned (line 319) that “this study also found that a smaller distance between drainage pipes lowered the N2O emissions…”, as a result of “… the greater probability of soluble nitrogen in the top-soil would be discharged along with the drainage water.”. If it happened in this experiment (drainage and nitrogen leaching were not measured), is this a viable consequence? Will be a solution to solve a problem by creating another?
Thank you for pointing out this. It is one deficient that we have mot monitored the nitrogen indicators in drainage water. The smaller distance between pipes resulted in the less N2O emissions; However, the greater risk of nitrogen loss and the higher cost will happen. Therefore, it is important to comprehensively consider possible factors and optimize the layout of hidden pipes through a multi indicator model. Thank you for your suggestion. We have added the discussion section of the article.
Additional comments
- line 11 – “Organic fertilizer application sand subsurface drainage are two important…”. Review the text.
Thank you for reminding. It has been corrected.
- line 17 “…50% organic fertilizer (50% OF), 25% organic fertilizer (25% OF) …” Review the text.
Yes, there should be no spaces in the abbreviations, and now we have removed them all. Thank you for your reminding.
- line 31 – “Keywords: N2O, organic fertilizer, subsurface drainage, saline soil, sunflower” With the exception of the word “sunflower”, all the others appear in the title. I suggest its replacement by other keywords
Thank you for your suggestion. We have completed all the replacement.
- line 207 – “Figure 1. Variation of N2O emission flux with days after transplanting …”. Transplanting?
Thank you for your reminding. It should be basal fertilization. We have revised the whole MS.

Reviewer 3 Report
This manuscript conducted by Zhai examined the impacts of organic-inorganic fertilizer application and subsurface drainage on N2O emissions in coastal saline soils cultivated with sunflowers. The rationale and objectives are clearly stated, and the experimental design using different drainage spacings and fertilizer mixes is appropriate to address the research questions posed. The results are thoroughly presented and discussed in the context of related literature. Overall, this is a well-written paper that makes a valuable contribution on an important topic. I have some suggestions below that I believe could further strengthen the quality of this publication.
Major Comments
In the Introduction, expand on prior related studies that have looked at N2O emissions in coastal saline soils specifically, to provide more context on the novelty of this work.
Provide more details in the Methods on the site history, soil characteristics, and agronomic management practices besides fertilization and drainage. This will aid interpretation of the results.
The discussion could be enhanced by more explicitly tying the observed trends back to proposed mechanisms such as changes in substrate availability, microbial activity, etc. Please consider these paper. Organic matter contributions to nitrous oxide emissions following nitrate addition are not proportional to substrate-induced soil carbon priming; Drying and rewetting cycles increased soil carbon dioxide rather than nitrous oxide emissions: A meta-analysis.
Conclusions could highlight the practical implications of these findings and give recommendations for how the drainage configuration and fertilization strategy could be optimized to lower N2O in this cropping system.
Minor Comments
Abstract: change "replacement" to "application" when referring to organic fertilizer
Be consistent in spelling subsurface as one or two words throughout
Add units when first mentioning 15 m, 20 m spacings
Figure 1: Change x-axis labels to be consistent for all graphs (e.g. use After Basal Fertilization)
Although the N2O emissions decreased and seems to be stable, while the accumulative emission were remain increase (Figs. 1-2), please give a couple of sentences to explain the reason that the measurement days were not longer than 120 day in the discussions.
Table 2: Add note specifying that S1, S2, S3 represent drainage spacings
Title: subscript of N2O.
Author information, I guess the full comma was used, please revise.
L12: Define N2O then use the abbreviation.
Figure 2: Add error bars.
Figure 3, y axis, leave a space between unit and title.
Overall, I found this to be a well-executed study generating useful insights on managing N2O emissions in coastal agriculture. Addressing the comments above would further improve the manuscript quality and clarity. I appreciate the opportunity to review this paper and believe it merits publication after revisions. Please feel free to contact me if you would like me to clarify or expand on any of my comments.
Moderate editing of English language required.
Author Response
- This manuscript conducted by Zhai examined the impacts of organic-inorganic fertilizer application and subsurface drainage on N2O emissions in coastal saline soils cultivated with sunflowers. The rationale and objectives are clearly stated, and the experimental design using different drainage spacings and fertilizer mixes is appropriate to address the research questions posed. The results are thoroughly presented and discussed in the context of related literature. Overall, this is a well-written paper that makes a valuable contribution on an important topic. I have some suggestions below that I believe could further strengthen the quality of this publication.
Thank you very much for your guidance and affirmation of our content.
- In the Introduction, expand on prior related studies that have looked at N2O emissions in coastal saline soils specifically, to provide more context on the novelty of this work.
Thank you for your valuable suggestion. Indeed, our innovation statement is not enough. One of our innovations was to study the synergistic effects of water conservancy, plants, and agronomic measures on N2O emissions. Therefore, we added: Is there an optimal coupling mode for water conservancy measure, plant cultivation, and fertilizer application that can ensure crop output while reducing N2O emissions, is worth studying.
- Provide more details in the Methods on the site history, soil characteristics, and agronomic management practices besides fertilization and drainage. This will aid interpretation of the results.
Thank you for your suggestion. We have supplemented the crops before this experiment, fertilization methods, irrigation methods, etc. previously planted in the experimental site, as you can see in the revised manuscript.
- The discussion could be enhanced by more explicitly tying the observed trends back to proposed mechanisms such as changes in substrate availability, microbial activity, etc. Please consider these paper. Organic matter contributions to nitrous oxide emissions following nitrate addition are not proportional to substrate-induced soil carbon priming; Drying and rewetting cycles increased soil carbon dioxide rather than nitrous oxide emissions: A meta-analysis.
Thank you very much for your comment. Indeed, these two articles are highly relevant to our content. Now, it has been added, as shown in literature 35-36.
- Conclusions could highlight the practical implications of these findings and give recommendations for how the drainage configuration and fertilization strategy could be optimized to lower N2O in this cropping system.
Thanks for your suggestion, we have made the following additions: We highlighted the important role of subsurface drainage, organic fertilizer application and sunflower planting for the sustainable development of coastal agriculture. We concluded that 25% organic fertilizer instead of inorganic fertilizer was beneficial for reducing N2O emissions while maintaining the crop yield. However, the spacing of drainage pipes should be determined based on a comprehensive consideration of N2O emissions, nitrogen loss, and engineering costs.
Minor Comments
- Abstract: change "replacement" to "application" when referring to organic fertilizer
Thank you for your suggestion. We have made two modifications, as shown in the revised Abstract.
- Be consistent in spelling subsurface as one or two words throughout
Thank you for your suggestion. Yes, subsurface is a word. The “-” between sub and surface may be caused by line wrap in shift scheduling.
- Add units when first mentioning 15 m, 20 m spacings
Thank you for your suggestion. We have added the units both in abstract and experimental design.
- Figure 1: Change x-axis labels to be consistent for all graphs (e.g. use After Basal Fertilization)
Thank you for your comment. We have modified the corresponding figures, as shown in Figure 1 and Figure 2, which have been replaced with After Basal Fertilization
- Although the N2O emissions decreased and seems to be stable, while the accumulative emission were remain increase (Figs. 1-2), please give a couple of sentences to explain the reason that the measurement days were not longer than 120 day in the discussions.
Thank you for your reminder. The crop growth season may be more representative. We added in the discussion: Our experiment observed N2O emissions during the 120 days of crop growth period, mainly due to that the frequent activities such as irrigation, drainage, and fertilization in this period have a more significant impact on N2O emissions.
- Table 2: Add note specifying that S1, S2, S3 represent drainage spacings
Thank you for your comment. We have added S1/S2/S3 to refer to the distance between concealed pipes and placed it at the end.
- Title: subscript of N2O.
Thank you for your reminding. We have revised it.
- Author information, I guess the full comma was used, please revise.
Thank you for your comment. We have re-entered the comma to ensure the format is correct
- L12: Define N2O then use the abbreviation.
Thank you for your reminding. We have added the nitrous oxide in both the introduction and abstract.
- Figure 2: Add error bars.
Thank you very much for your comment. Indeed, the standard deviation was omitted, and we have already supplemented it
- Figure 3, y axis, leave a space between unit and title.
Thank you for your reminding. We have added the blank space.
- Overall, I found this to be a well-executed study generating useful insights on managing N2O emissions in coastal agriculture. Addressing the comments above would further improve the manuscript quality and clarity. I appreciate the opportunity to review this paper and believe it merits publication after revisions. Please feel free to contact me if you would like me to clarify or expand on any of my comments.
Thank you very much for your guidance on our experiment. Your valuable comment has been very helpful to us.

Round 2
Reviewer 1 Report
Thank you for incorporating all the revisions.
Reviewer 2 Report
The manuscript was improved and the authors considered the suggestions made.